# The Recycling of End-of-Life Lithium-Ion Batteries and the Phase Characterisation of Black Mass

Laurance Donnelly [1,*], Duncan Pirrie [2], Matthew Power [3], Ian Corfe [4], Jukka Kuva [4], Sari Lukkari [4], Yann Lahaye [4], Xuan Liu [4], Quentin Dehaine [4], Ester M. Jolis [4] and Alan Butcher [4]

1    Alfred H Knight International, Kings Business Park, Kings Drive, Prescot L34 1J, UK
2    Faculty of Computing, Engineering and Science, University of South Wales, Pontypridd CF37 4AD, UK
3    Vidence Inc., 213L 4288 Lozells Avenue, Burnaby, BC V5A 0C7, Canada
4    Geological Survey of Finland (GTK), Vuorimiehentie 5, P.O. Box 96, FI-02151 Espoo, Finland
*    Correspondence: laurance.donnelly@ahkgroup.com; Tel.: +44-(0)-151-482-4276

**Abstract:** Black mass is the industry term applied to end-of-life (EoL) lithium-ion batteries that have been mechanically processed for potential use as a recycled material to recover the valuable metals present, including cobalt, lithium, manganese, nickel and copper. A significant challenge to the effective processing of black mass is the complexity of the feed material. Two samples of black mass from a European source were analysed using a combination of methods including automated SEM-EDS (AMICS) to characterise and quantify the phases present and particle chemistry. Micro X-CT imaging, overlain onto automated mineralogy images, enabled the 3D morphology of the particles to be determined. Micro-XRF was used to map the copper, nickel, manganese and cobalt-bearing phases. Since Li cannot be detected using SEM-EDS, its abundance was semi-quantified using laser ablation inductively coupled plasma mass spectrometry (LA-ICP-MS). The integration of these complimentary analytical methods allowed for detailed phase characterisation, which may guide the potential hydrometallurgical or pyrometallurgical recycling routes and chemical assaying.

**Keywords:** black mass; lithium-ion batteries; EoL batteries; sampling; automated mineralogy; SEM-EDS; Micro X-CT; Micro-XRF; LA-ICP-MS





## 1. Introduction

End-of-life (EoL) lithium-ion batteries may be recycled to recover valuable metals. Following the removal of residual electrolyte, the batteries undergo a physical, thermo-mechanical process to produce a fine powder known as 'black mass' [1–6]. The volume of black mass is anticipated to increase over the next few decades as more batteries come onto the market in electrified transportation and stationary storage, and reach EoL [7,8]. If there is a future scarcity of primary geological sources of battery minerals, the recycling of black mass could provide an alternative supply of battery metals, including cobalt, lithium, manganese, nickel and copper, whilst also reducing the need for waste disposal. This source has been estimated to partially make up for shortfalls in battery minerals between demand and production as demand rises [9]. Black mass is an important feedstock for subsequent hydrometallurgical or pyrometallurgical processing to recover the elemental metals of value [10–19]. However, to date limited research has been conducted and published on the characterisation of black mass [20,21].

The objective of this paper is to provide the results of an investigation to determine the principal phase characteristics of a sample of black mass from a source in Europe. Some of the preliminary results were presented at sampling conferences throughout Europe [22–24] and published in non-peer review conference proceedings [25] and an industry magazine [26]. However, the aim of this manuscript is to present how, through the integration of a range of analytical workflows, black mass materials can be fully characterised as a first step in their reprocessing and the recovery of metals from this waste stream.

## 2. Overview of Analytical Methods

In this study, we provide a methodology for the detailed characterization of black mass through the integration of a range of analytical approaches. A protocol will be important for future processing studies and also for companies trading in black mass. The sample of European black mass was analysed using:

- Manual scanning electron microscopy (SEM), with linked energy dispersive spectrometers (EDS).
- Automated SEM-EDS.
- Micro X-ray computed tomography (X-CT).
- Scanning Micro-XRF elemental and phase analysis (Micro-XRF).
- Laser ablation inductively coupled plasma mass spectrometry (LA-ICP-MS).

### 2.1. SEM-EDS

Preliminary SEM images were collected using a TESCAN TIMA-X GMU field emission gun scanning electron microscope. Subsequent analysis was carried out using a JEOL JSM-7100F field emission scanning electron microscope.

### 2.2. Automated Phase Analysis

Automated phase analysis is based on a scanning electron microscope with linked energy dispersive spectrometers (SEM-EDS). The analysis provides rapid determination and quantification of the mineralogy/phase chemistry, particle size and shape of a variety of sample types. Data collection is operator-independent, with the acquisition of very large data sets. As such, the results are statistically reliable and provide highly reproducible analyses [27].

Analysis was undertaken with a Hitachi SU3900 scanning electron microscope fitted with a single large area (60 mm$^2$) Bruker silicon drift detector (SDD) energy dispersive spectrometer and running the AMICS automated mineralogy package. Beam conditions were optimised for analysis and, therefore, an accelerating voltage of 20 kV coupled with a beam current of approximately 15 nA was used.

### 2.3. Micro X-CT

Analysis was carried out using a GE Phoenix v | tome | x s 240 Micro X-ray computed tomography system. The components in the samples, such as mineral phases, were identified using other methods, for example SEM-EDS, from a two-dimensional surface. The information from this identification was then applied to the density map. The same measured epoxy block sample was used for Micro X-CT as had initially been imaged using AMICS, interactively analysed using SEM-EDS and finally 3D-imaged using Micro X-CT. This same sample was also used for the LA-ICP-MS analysis. In total, 2700 angle step projection images were taken during the scan. For each angle, the detector waited a single exposure time and then took an average of three exposures, with each single exposure time being 1000 ms. This resulted in a total scan time of 3 h, with a voltage of 100 kV, current 50 mA and voxel size 5 μm. Reconstruction was performed using GE phoenix datos / x 2 reconstruction software version 2.6.1 and data analysed and images constructed using the Thermo Fisher PerGeos® software version 2020.2.

### 2.4. Micro-XRF

Micro X-Ray Fluorescence (Micro-XRF) mapping was carried out using a Bruker M4 Tornado Plus AMICS scanning micro-XRF. The instrument uses Rh X-ray 30-Watt Rh anode target, two simultaneously operating 30 mm$^2$ XFlash® silicon drift detectors (SDDs) with an energy resolution of <145 eV (measured on MnKα) at 275 kcps. The accelerating voltage was 50 kV with a beam current of 600 μA, using a fixed spot size of 20 μm under 2 mbar vacuum. A standard integration time of 10 ms per pixel and a spacing of 10 μm were employed for the analysis. The qualitative elemental maps were generated using the Bruker M4 software (version 1.6.621.0).

### 2.5. LA-ICP-MS

Laser ablation inductively coupled mass spectrometry (LA-ICP-MS) analyses of the same black mass sample were carried out using a Nu AttoM SC-ICPMS (Nu Instruments Ltd., Wrexham, UK) and a 193 nm Excimer ArF laser-ablation system (Photon Machines, San Diego, CA, USA). The laser was run at a pulse frequency of 5 Hz and a pulse energy of 5 mJ at 30% laser energy to produce a fluence of 2.17 J/cm$^2$ on the sample surface with a 20 μm spot size. Each analysis was initiated with a 20 s baseline measurement followed by switching on the laser for 90 s for signal acquisition. Analyses were made using time-resolved analysis (TRA) with continuous acquisition of data for each point (generally following the scheme of primary standard, quality control standard, 10–20 unknowns). The NIST glass 610 and 612 were used for external standardisation. Aluminium and cobalt were used as the internal standards depending on the closeness of concentration between NIST 610 and the samples. The measurements were performed on 75 isotopes covering 68 elements at low mass resolution ($\Delta M/M = 300$) using the 'Fastscan' mode. Data reduction was handled using a GLITTER4.4.4 excel spreadsheet, which allowed the baseline subtraction, outlier removal, the integration of the signal over a selected time resolve area and the quantification.

## 3. Phase Characterisation

### 3.1. Visual Examination and Binocular Microscopy

Two subsamples of black mass powder were visually inspected using binocular microscopy for preliminary phase characterisation and comprised a 100 g unprepared sample and a 60 g prepared sample. The difference between the two samples was that the prepared sample was crushed and ground to a less than 180 μm powder. As the analysis was aimed to understand the components present in a black mass sample, most of the analyses focused on the coarser-grained unprepared sample (Figure 1).

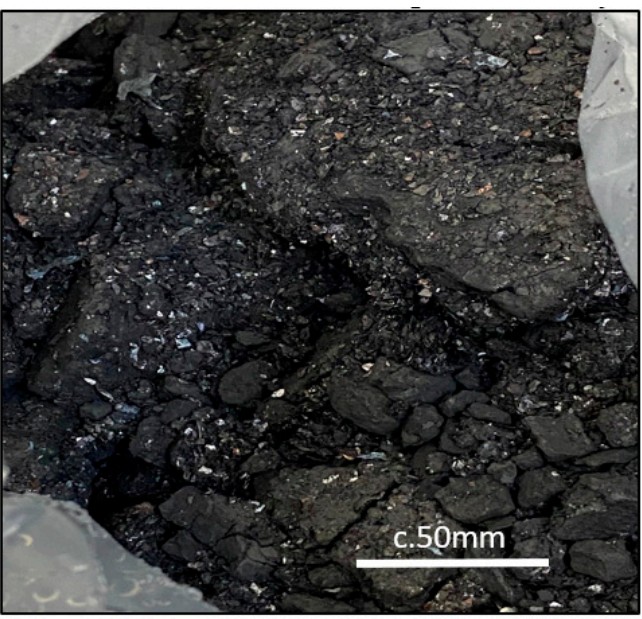

**Figure 1.** Sample of black mass from a European source submitted for analysis. Note the complex heterogeneous texture of the black mass and a coarser oversized fraction.

### 3.2. Sample Preparation

Prior to automated analysis, two approximately 1 g subsamples of both the prepared and unprepared samples were taken and mixed in individual rectangular moulds with epoxy resin and left to cure for 24 h as replicates A and B. Both replicates for both samples were then cut in half and remounted in new moulds so that the cross-section view through

both halves of the subsample could be imaged. The two blocks were then labelled, backfilled with resin and cured for two hours at 40 °C. The rationale for doing this was because in mineral processing applications, graphite is usually added to the mineral commodity prior to analysis to minimise any effects of differential particle settling within the blocks. As graphite is a component of the LIB black mass, it could not be added to the samples, and as such, the cross-section view was prepared and measured so that any differential particle setting could be visualised. The surface of the blocks was polished and carbon coated prior to analysis.

### 3.3. Manual SEM

The bulk-powdered samples were imaged using manual SEM. The analysis showed that the powders were composed of a wide range of particle sizes and shapes. Larger particles were commonly coated with finer particles, and the larger particles were generally composite in nature (Figure 2).

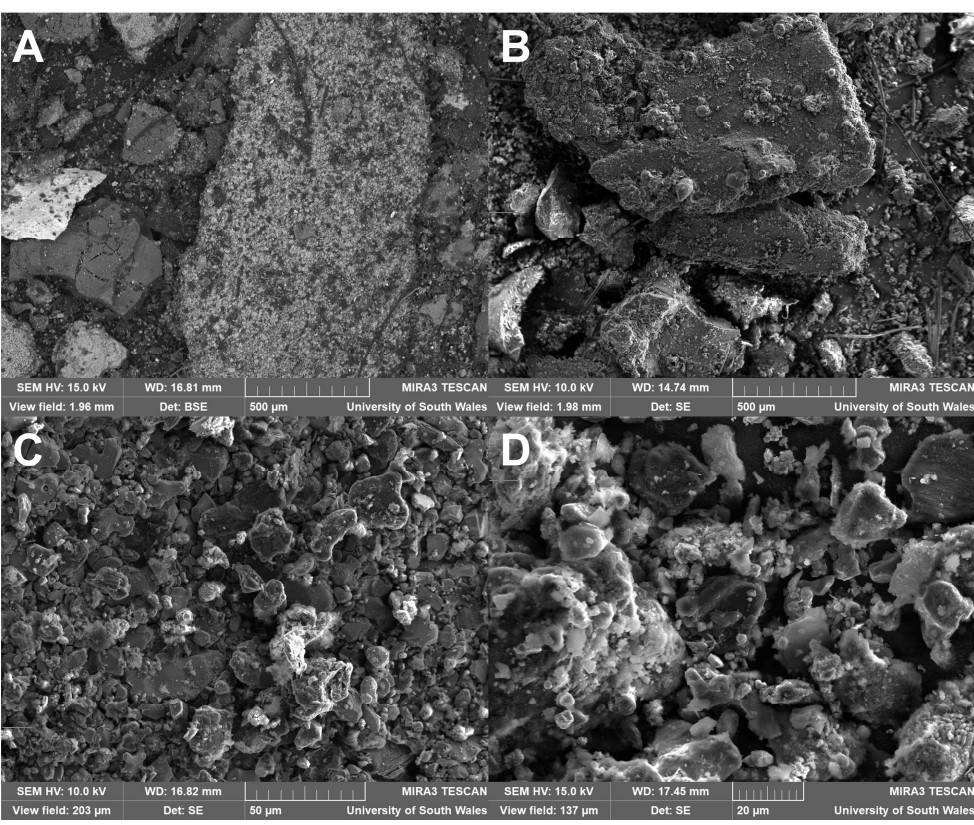

**Figure 2.** (**A**–**D**) Scanning electron microscope images of the unprepared black mass sample.

### 3.4. Automated Phase Analysis

The samples were measured using the segmented field image mode of analysis. This analytical mode subdivided the BSE image into domains (segments) of similar brightness, which represent different composition particles, and then acquired a representative EDS X-ray spectrum from a point within the segment. The phase identified was then assigned to the entire segment. Measurements were optimised to highlight both textural and modal abundance information, and so an effective image resolution of 1.6 μm was used.

The EDS spectra acquired during the measurements were compared with a library of measured and synthetic standards and a phase identification was made on a closest match basis. Phases which were not represented in the standards list at the time of measurement were added either by acquiring reference spectra directly from the sample, or by creating a reference spectrum from the measurement itself. As the standards list can comprise hundreds of reference spectra, the data were grouped into a final, more manageable,

reported list of phases. A table outlining which compositional groups were included in each of the reported groups is provided in Appendix A. During AMICS analysis, a full SEM-BSE map of the area imaged was also collected.

Three measurements were carried out on both the unprepared and the prepared subsamples:

- Replicate block A—An area including the two cross-sectioned halves in the block (100× magnification/1.6 μm measurement resolution).
- Replicate block B—An elongated area of one of the sectioned halves (100× magnification/1.6 μm measurement resolution).
- A detailed area of one of the halves (150× magnification/0.83 μm measurement resolution).

The raw data were then processed, and each analysis point was assigned to a compositional grouping. This was initially carried out based on the measured chemical spectra, without consideration of what component/structural element of the LIB the particle was probably derived from. This approach was used so that the data were not manipulated to match expected compositional groups. The definitions of the compositional groups are provided in Appendix A. The analytical results are presented in Appendix B. There was slightly greater variance between the three measurements in the 'unprepared' sample when compared with the 'prepared' sample (Figure 3). This was interpreted as a function of the grain size difference between the two samples; in the coarser-grained sample, the particles were larger, and as such, there was greater analytical variability between the measured samples, based on the subset of particles measured. This effect was minimised in the prepared (ground) sample, but with the reduction in particle size, textural attributes were lost, making phase assignment to battery components more challenging. There was generally good agreement between the different subsamples. There was also no significant variation between the two resolution measurements.

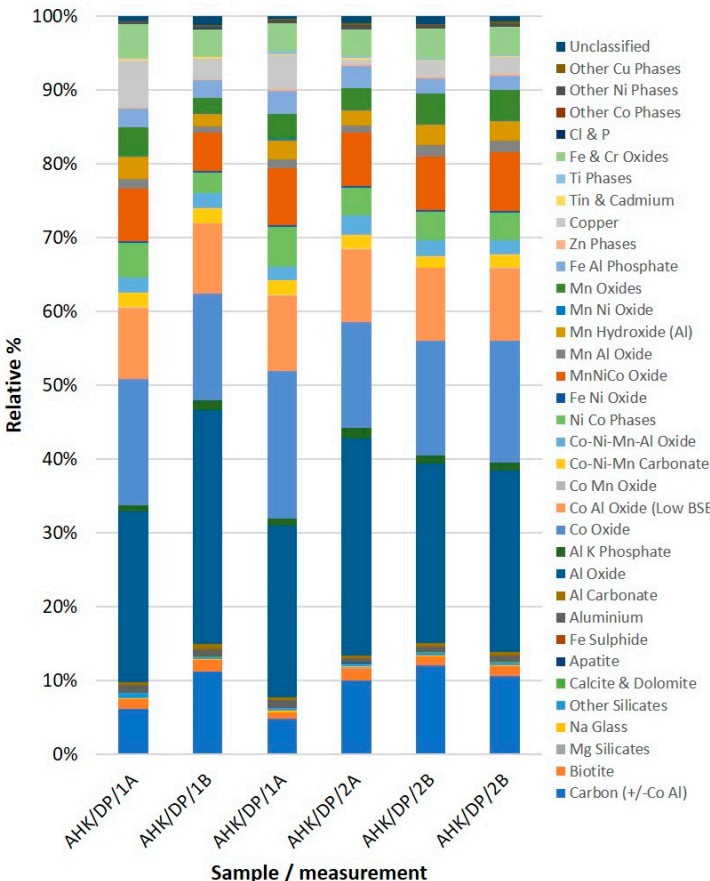

**Figure 3.** Black mass sample automated SEM-EDS phase abundance. AHK/DP/1 was the 'unprepared' sample, AHK/DP/2 was the 'prepared' (i.e., milled) sample.

These compositional groups can be reclassified in terms of the likely attribution of the particles to the original battery components: (a) graphite, (b) aluminium foil, (c) copper foil and (d) lithium metal oxides. The lithium metal oxide category can be further divided into the five main cathode types, based on their chemistry: (i) NMC (LiNiMnCo oxides), (ii) LCO (LiCo oxides), (iii) LM(N)O (LiMn(Ni) oxides), (iv) LFP (LiFe phosphates) and (v) NCA (LiNiCoAl oxides). Within the data there were also (a) particles currently assigned to mineralogical categories; these may have been derived from the batteries or cross contamination with environmental materials, and (b) metal phases, the attribution of which was unclear. These categories are shown in Table 1. The relative abundances of the different phases grouped into the different commodity categories are summarised in Table 2 and are shown graphically in Figure 4.

**Table 1.** Automated SEM-EDS analysis results for graphite, aluminium foil, lithium metal oxides (cathodes), mineral phases and unattributed metal phases. LMO stands for Li Metal Oxides, other abbreviations are given in the text.

| Component (Cathode Type) | Sub-Phase | Sample | | | | | | |
|---|---|---|---|---|---|---|---|---|
| | | AHK/DP/1 | AHK/DP/1 | AHK/DP/1 | AHK/DP/2 | AHK/DP/2 | AHK/DP/2 | Mean |
| Graphite | Carbon (+/−Co,Al) | 6.17 | 11.21 | 4.89 | 10.08 | 12.05 | 10.62 | 9.17 |
| Aluminium Foil | Aluminium | 1.07 | 1.00 | 1.08 | 0.69 | 0.70 | 0.84 | 0.90 |
| | Al Carbonate | 0.40 | 0.65 | 0.35 | 0.27 | 0.49 | 0.56 | 0.45 |
| | Al Oxide | 23.05 | 31.68 | 23.20 | 29.42 | 24.33 | 24.52 | 26.03 |
| | Al K Phosphate | 0.94 | 1.36 | 1.05 | 1.43 | 1.15 | 1.11 | 1.17 |
| Copper Foil | Copper | 6.32 | 2.73 | 4.59 | 0.64 | 2.03 | 2.28 | 3.10 |
| | Other Cu Phases | 0.08 | 0.07 | 0.13 | 0.19 | 0.18 | 0.22 | 0.14 |
| LMO (LCO) | Co Oxide | 17.08 | 14.46 | 19.97 | 14.33 | 15.49 | 16.45 | 16.30 |
| | Co Al Oxide (Low BSE) | 9.52 | 9.42 | 10.12 | 9.78 | 9.84 | 9.83 | 9.75 |
| | Other Co Phases | 0.01 | 0.01 | 0.01 | 0.00 | 0.02 | 0.05 | 0.02 |
| LMO (NMC) | Co Mn Oxide | 0.05 | 0.06 | 0.04 | 0.07 | 0.06 | 0.05 | 0.06 |
| | Co-Ni-Mn Carbonate | 2.01 | 2.00 | 2.09 | 1.94 | 1.60 | 1.78 | 1.90 |
| | Co-Ni-Mn-Al Oxide | 2.10 | 2.07 | 1.87 | 2.65 | 2.09 | 2.01 | 2.13 |
| | Ni Co Phases | 4.69 | 2.77 | 5.37 | 3.72 | 3.88 | 3.68 | 4.02 |
| | Fe Ni Oxide | 0.26 | 0.23 | 0.24 | 0.25 | 0.25 | 0.23 | 0.24 |
| | MnNiCo Oxide | 7.10 | 5.16 | 7.74 | 7.27 | 7.21 | 7.94 | 7.07 |
| | Mn Ni Oxide | 0.15 | 0.08 | 0.24 | 0.08 | 0.08 | 0.10 | 0.12 |
| | Other Ni Phases | 0.47 | 0.46 | 0.42 | 0.56 | 0.49 | 0.45 | 0.48 |
| LMO (LFP) | Fe Al Phosphate | 2.46 | 2.32 | 3.13 | 3.06 | 2.09 | 1.93 | 2.50 |
| LMO (LM(N)O) | Mn Al Oxide | 1.32 | 0.94 | 1.20 | 0.98 | 1.61 | 1.59 | 1.27 |
| | Mn Hydroxide (Al) | 3.02 | 1.64 | 2.53 | 2.00 | 2.75 | 2.71 | 2.44 |
| | Mn Oxides | 3.85 | 2.13 | 3.40 | 2.95 | 4.06 | 4.11 | 3.42 |
| Mineral Phases | Biotite | 1.35 | 1.56 | 0.84 | 1.58 | 1.21 | 1.32 | 1.31 |
| | Mg Silicates | 0.03 | 0.09 | 0.03 | 0.17 | 0.06 | 0.06 | 0.07 |
| | Na Glass | 0.13 | 0.08 | 0.13 | 0.12 | 0.10 | 0.07 | 0.10 |
| | Other Silicates | 0.62 | 0.25 | 0.36 | 0.37 | 0.32 | 0.42 | 0.39 |
| | Calcite and Dolomite | 0.04 | 0.07 | 0.05 | 0.08 | 0.11 | 0.03 | 0.06 |
| | Apatite | 0.00 | 0.03 | 0.00 | 0.01 | 0.00 | 0.00 | 0.01 |
| | Fe Sulphide | 0.02 | 0.00 | 0.02 | 0.00 | 0.00 | 0.00 | 0.01 |
| | Cl and P | 0.02 | 0.04 | 0.04 | 0.02 | 0.01 | 0.01 | 0.02 |
| | Unclassified | 0.56 | 1.22 | 0.38 | 1.03 | 1.04 | 0.77 | 0.83 |
| Unattributed Metal Phases | Zn Phases | 0.20 | 0.14 | 0.16 | 0.16 | 0.23 | 0.21 | 0.18 |
| | Tin and Cadmium | 0.29 | 0.24 | 0.18 | 0.23 | 0.23 | 0.17 | 0.22 |
| | Ti Phases | 0.13 | 0.11 | 0.28 | 0.18 | 0.14 | 0.05 | 0.15 |
| | Fe and Cr Oxides | 4.50 | 3.70 | 3.91 | 3.68 | 4.11 | 3.86 | 3.96 |

**Table 2.** Summary of automated SEM-EDS analysis results with particles assigned to battery component categories.

| Phase | AHK/DP/1 Mean | AHK/DP/2 Mean | Combined |
|---|---|---|---|
| Graphite | 7.42 | 10.92 | 9.17 |
| Cu foil | 4.63 | 1.85 | 3.24 |
| Al foil | 28.61 | 28.50 | 28.56 |
| LMO cathode | 52.07 | 51.35 | 51.71 |
| Other metals | 4.61 | 4.41 | 4.51 |
| Minerals | 2.65 | 2.97 | 2.81 |

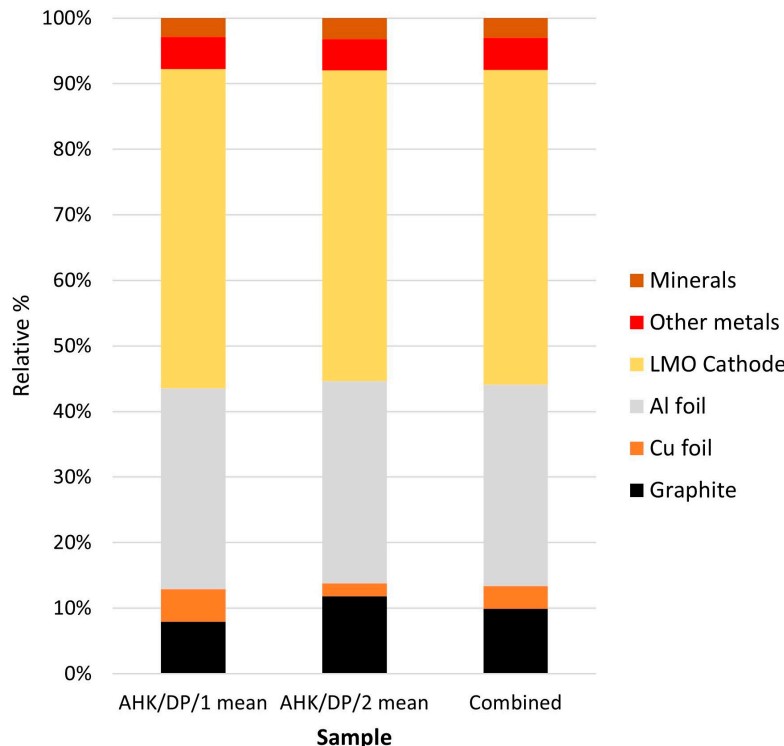

**Figure 4.** Black mass sample automated SEM-EDS relative abundance of different battery components.

There was a good correspondence between the two subsamples analysed, with an average composition of the sample being 9.2% graphite, 3.2% copper foil, 28.6% aluminium foil, 51.7% lithium metal oxides, 4.5% other metals and 2.8% mineral phases. In general, these results are broadly comparable with the other analysis, whose sample comprised: 17.5% graphite, 47.4% lithium metal oxides (NMC and LCO), 9.5% aluminium foils, 3.4% copper foils and 15.2% alloys [20]. Black mass samples are likely to be heterogeneous, so a strong correspondence is not necessarily likely. Key differences were in the relative abundance of aluminium foil, which was much more abundant in the analysis reported here; less abundant graphite and alloys, although if the 'other metals' category was assumed to be alloy components, then these phases were of similar abundance. The compositional groups assumed to relate to the cathode components are shown in Table 2 along with the assigned cathode type. In the sample analysed, particle types interpreted as derived from the cathodes were dominated by three cathode types: LCO, NMC and minor LFP and LM(N)O. The relative abundance of (a) all particle types assumed to be derived from the cathodes, and (b) the relative abundance of particles assigned to LCO, NMC, LFP and LM(N)O are shown graphically in Figures 5 and 6 for the three replicate analyses per sample.

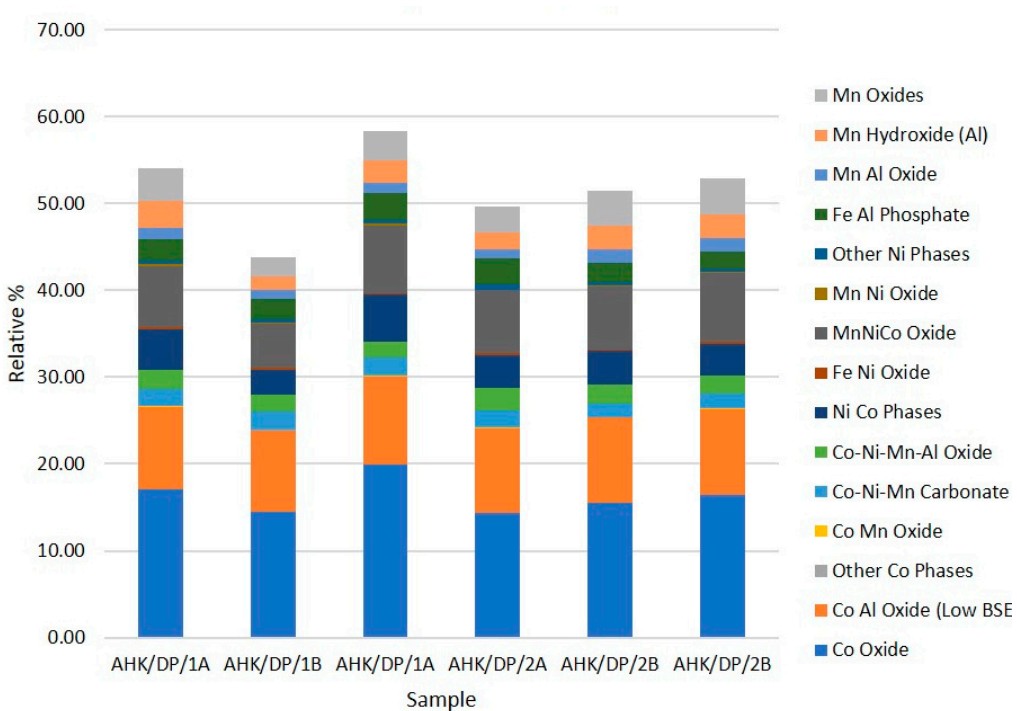

**Figure 5.** Relative abundance of compositional groups derived from cathode components in the analysed subsamples.

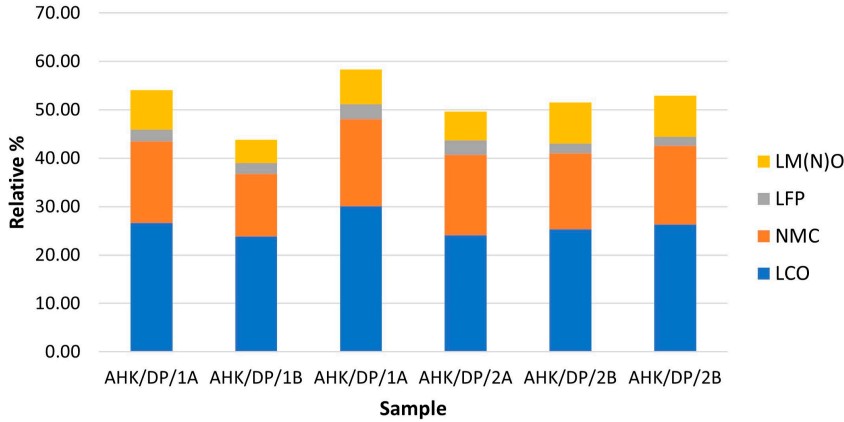

**Figure 6.** Relative abundance the cathode compositional groups assigned to LCO, NMC and LFP in the analysed subsamples.

There was, in general, a good correspondence between the subsamples analysed. The outlier in the data was the analysis for the unprepared sample analysed at a higher resolution. The increased resolution resulted in a smaller area being analysed and potentially a reduction in the representativeness of the analysis. Once again, the results were comparable with other investigations, with the sample analysed herein being composed of 50.4% LCO, 31% NMC, 4.8% LFP and 13.8% LM(N)O, whilst the other sample analysed (2021) comprised 66.4% LCO and 33.6% NMC [20].

During automated SEM-EDS analysis, an image of the measured sample was also generated, with each phase shown as a separate colour. These images provided textural information regarding the particles present within the sample. Figure 7 shows the automated analysis and corresponding SEM-BSE image for subsample 1A of the unprepared sample. The images show that there was no preferred particle segregation as a result of mounting the sample in resin. In the prepared (ground) sample, the particle textures were lost, as particle size was reduced. As the ground sample was no longer representative

of the original material, this approach is not recommended as, for example, the leaching characteristics of a black mass sample may in part be a function of whether or not the different battery components have been successfully separated (liberated) during crushing and grinding in the recycling facility. Grinding experiments could, however, be carried out to determine the particle size at which optimum liberation is achieved, and this could be quantified through automated SEM-EDS analysis.

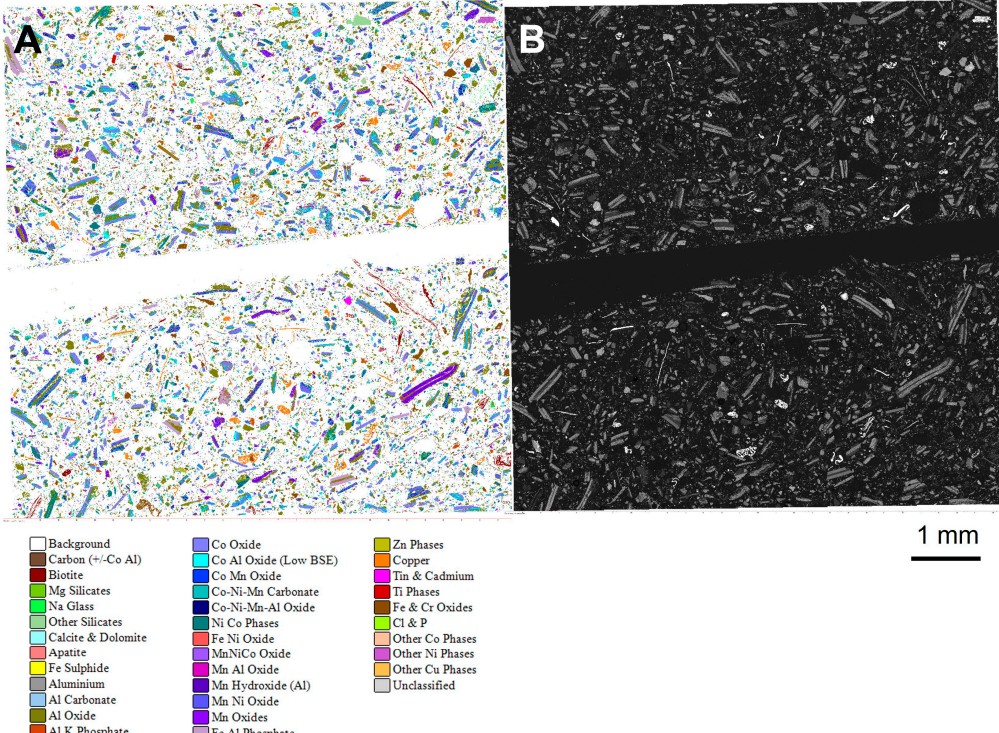

**Figure 7.** (**A**) Automated mineralogy and (**B**) SEM-BSE images for sample AHK/DP/1 from the unprepared sample, subsample 1A.

The automated mineralogy images allowed the particle types, as seen in the 2D measured sections, to be visualised. The majority of the larger particles were mixed phases, often with the aluminium foils coated with the different lithium metal oxide compounds. The copper foil was predominantly a separate phase, with an appearance in the 2D faces similar to that of a copper wire. However, as shown by the 3D Micro X-CT imaging, these are thin and complexly folded copper sheets. Texturally, the particles were complex, but the imaging allowed them to be distinguished. In the aluminium foils with lithium metal oxide coatings, the organic binding agents appeared dark grey in the BSE images and were not characterised in the automated SEM-EDS analysis. Aluminium oxides associated with LFP (lithium Fe phosphate) were also texturally distinctive (Figure 8).

### 3.5. Interactive SEM-EDS Analysis

Interactive SEM-EDS spot analysis was carried out to test the automated SEM-EDS particle classifications, and to gather additional, more detailed chemical analyses from the different phases identified. The interactive analysis confirmed the compositional groupings achieved through the automated SEM-EDS analysis, but also provided additional data regarding the detailed composition and variability of the different phases/battery components identified. The automated SEM-EDS AMICS image, SEM-BSE and detailed spot analyses for copper metal also showed this to contain trace cobalt (Figure 9).

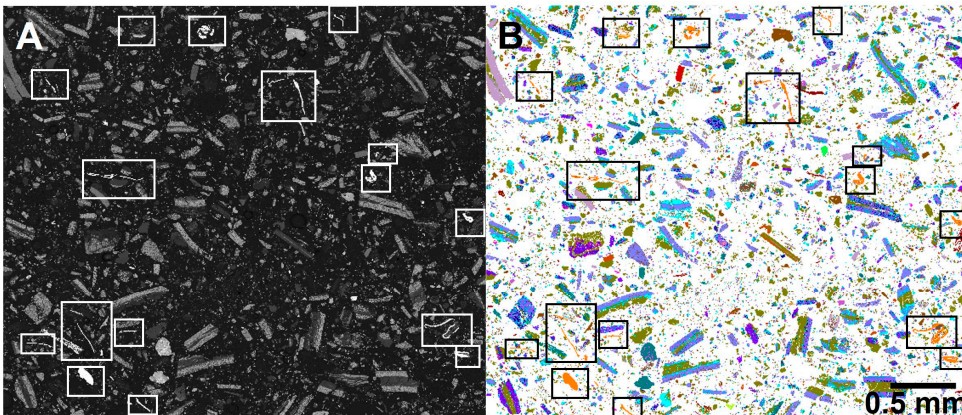

**Figure 8.** (**A**) SEM-BSE and (**B**) automated mineralogy images for the unprepared sample. The dominant large particle types are aluminium foils with a range of lithium metal oxide coatings. The bright phase in the SEM-BSE image, shown orange in the automated mineralogy image and highlighted within the boxes, is copper foil, which in 2D sections looks like folded wire.

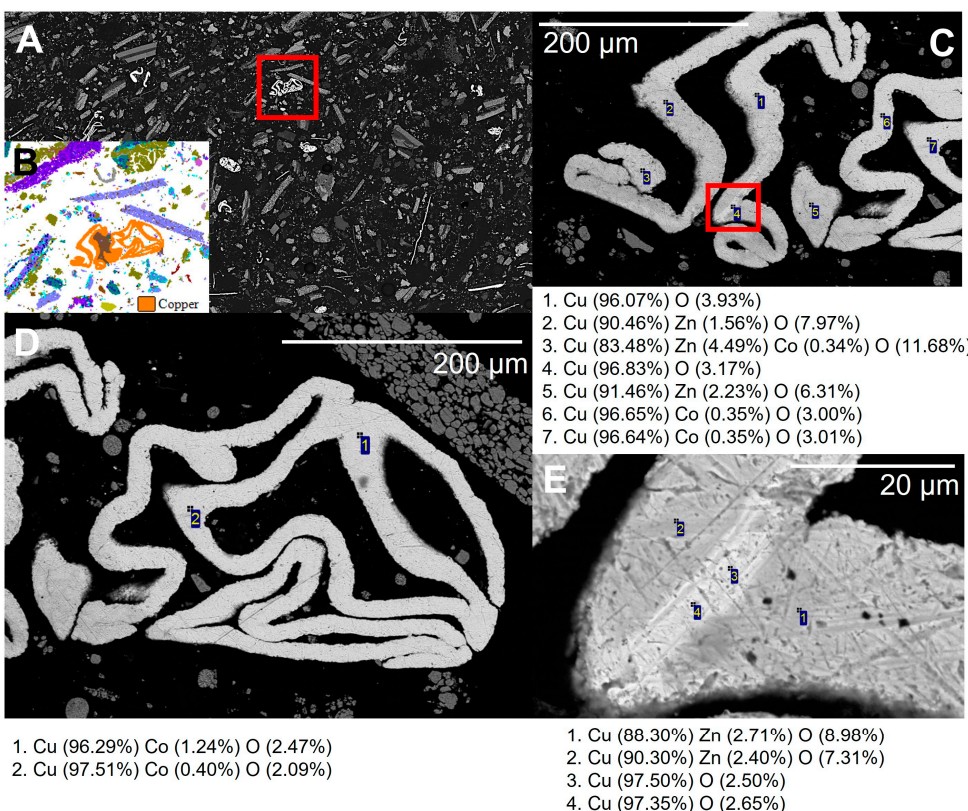

1. Cu (96.07%) O (3.93%)
2. Cu (90.46%) Zn (1.56%) O (7.97%)
3. Cu (83.48%) Zn (4.49%) Co (0.34%) O (11.68%)
4. Cu (96.83%) O (3.17%)
5. Cu (91.46%) Zn (2.23%) O (6.31%)
6. Cu (96.65%) Co (0.35%) O (3.00%)
7. Cu (96.64%) Co (0.35%) O (3.01%)

1. Cu (96.29%) Co (1.24%) O (2.47%)
2. Cu (97.51%) Co (0.40%) O (2.09%)

1. Cu (88.30%) Zn (2.71%) O (8.98%)
2. Cu (90.30%) Zn (2.40%) O (7.31%)
3. Cu (97.50%) O (2.50%)
4. Cu (97.35%) O (2.65%)

**Figure 9.** (**A**) SEM-BSE image and (**B**) automated mineralogy (AMICS image) from the unprepared sample, subsample showing a copper foil particle. (**C**–**E**) SEM-BSE images and spot EDS analysis shows the presence of minor cobalt and zinc within the copper metal.

In most particles, the aluminium oxide foils were inter-layered with the lithium metal oxides. In some cases, crushing and grinding liberated the aluminium oxide foils from the lithium metal oxides. Spot chemical analysis showed that these were aluminium-rich, with trace levels of cobalt and palladium (Figure 10). Representative particles assigned to the cathode LMO and NMC categories showed that the cathode particles were the dominant large grains within the analysed sample. Particles assigned to the LMO cathode category showed that the MnO was relatively pure, other than trace magnesium and aluminium

(Figure 11). Particles assigned to the NMC cathode category (LiNiMnCo oxides) and associated spot chemical analyses showed significant variance in the ratios of nickel, cobalt and manganese, as reflected in the number of compositional groups within the automated SEM-EDS particle characterisation.

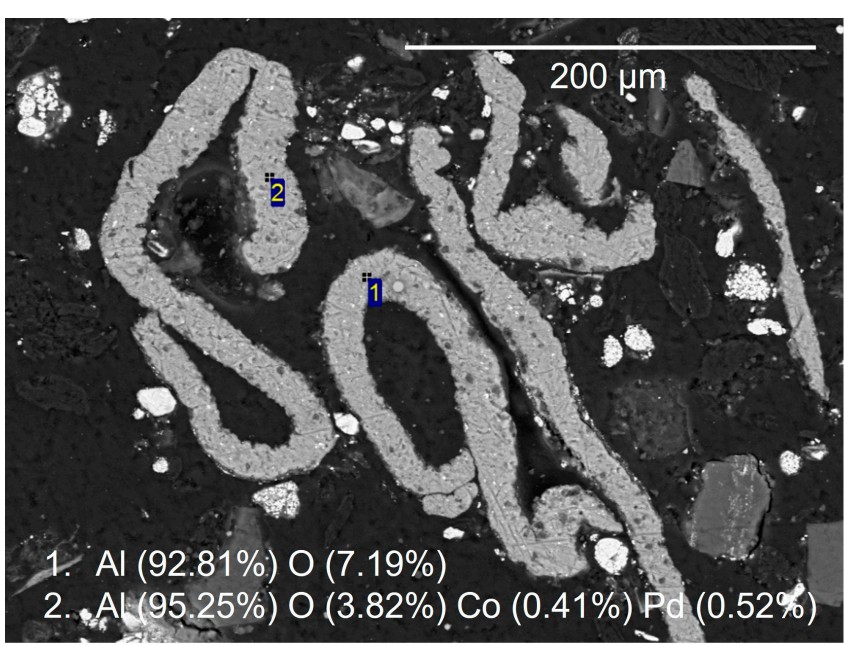

1. Al (92.81%) O (7.19%)
2. Al (95.25%) O (3.82%) Co (0.41%) Pd (0.52%)

**Figure 10.** SEM-BSE images from the unprepared sample, showing an aluminium, foil particle. Spot EDS analysis shows the presence of minor zinc, cobalt and palladium within the copper metal.

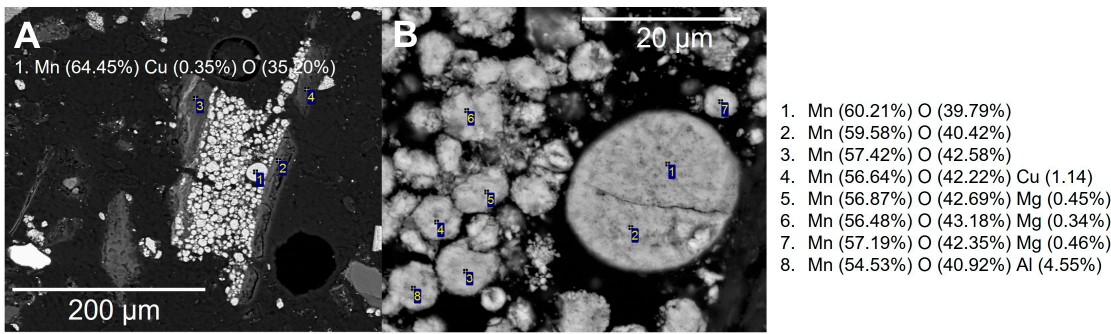

1. Mn (60.21%) O (39.79%)
2. Mn (59.58%) O (40.42%)
3. Mn (57.42%) O (42.58%)
4. Mn (56.64%) O (42.22%) Cu (1.14)
5. Mn (56.87%) O (42.69%) Mg (0.45%)
6. Mn (56.48%) O (43.18%) Mg (0.34%)
7. Mn (57.19%) O (42.35%) Mg (0.46%)
8. Mn (54.53%) O (40.92%) Al (4.55%)

A. 1. Mn (64.45%) Cu (0.35%) O (35.20%)

**Figure 11.** (**A**) SEM-BSE image of MnO particles associated with aluminium foils assigned to the LMO cathode category. (**B**) SEM-BSE image of MnO particle and spot EDS analyse.

### 3.6. Micro X-CT

On completion of the interactive SEM-EDS analysis, the same sample block from the unprepared sample was imaged using Micro X-CT scanning [28]. The aim of the Micro X-CT analysis was to image the 3D form of the particles. The automated SEM-EDS and interactive SEM-EDS data were all derived from the analysis of the 2D polished face of the analysed sample. The automated SEM-EDS analyses were overlain on the surface of the Micro X-CT scan to allow correlation between the 2D and 3D imaging. The results of the Micro X-CT imaging are also available as separate video files. When viewed at an oblique angle, the 3D structure of the black mass particles can be distinguished. When a fragment of copper foil is viewed, it can be visualised that the wire-like form of the copper foil as seen in the 2D view is a cross section through a complexly folded copper foil sheet in 3D (Figure 12). In contrast, there were a number of cathode sheets with aluminium foil coated

by lithium metal oxides. In 2D, these are rectangular in form, whilst the 3D imaging shows that they are platy sheets in geometry.

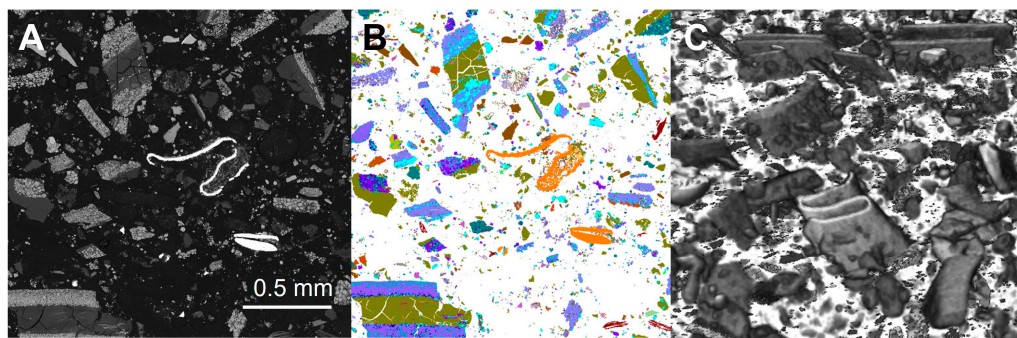

**Figure 12.** (**A**) SEM-BSE and (**B**) automated mineralogy (AMICS) imaging showing 2D section through a copper foil particle (bright grey phase in image A, orange in image B. (**C**) There particles are folded Cu sheets when imaged in 3D when using Micro X-CT (**C**). 0.5 mm scale bar applies to all images.

### 3.7. Micro-XRF

The same polished blocks used for automated mineralogy, Micro X-CT and LA-ICP-MS were re-analysed using an M4 scanning Micro-XRF [29]. The aim of the Micro XRF analysis was to reveal the main components of the black mass samples in terms of the metals of potential economic value, including copper, cobalt, nickel, manganese, zinc, nickel and phosphorus. These maps were also found to be useful in navigating areas of interest in the manual SEM and for comparison with the automated mineralogy results (Figure 13).

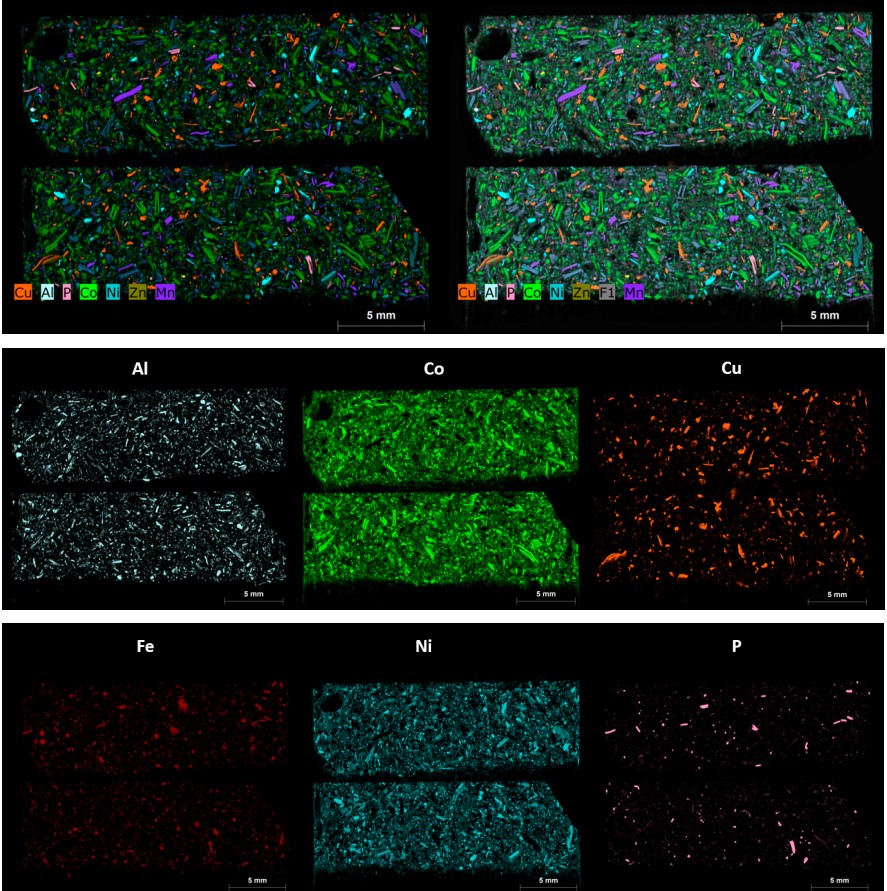

**Figure 13.** *Cont*.

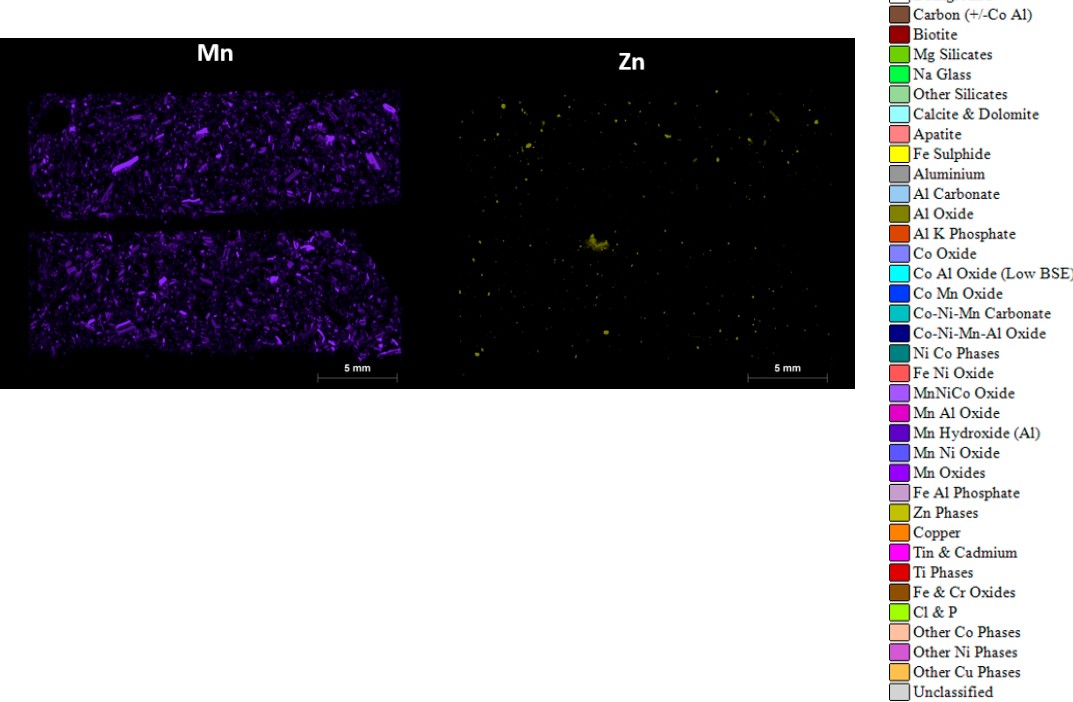

**Figure 13.** Micro-XRF combined and separate elemental intensity maps for sample AHK/DP/1, subsample 1A, displaying the various particle types and their phase compositions, in terms of aluminium, (Al), cobalt (Co), copper (Cu), iron (Fe), nickel (Ni), phosphorous (P), manganese (Mn) and zinc (Zn). The colour scheme used equates to the same colour codes used for the automated mineralogy. The scale bar shown on each image is 5 mm.

## 4. Detection and Quantification of Lithium

On completion of the interactive Micro X-CT, selected phases were analysed for minor and trace element analysis using LA-ICP-MS. Sixty-eight elements including lithium were measured. Elemental concentrations were calculated based on a 'non-matrix matched' silicate glass standard. Spot analyses (20 μm diameter size) from specific phases rather than transects were conducted. These were all particles interpreted as cathode fragments, in which lithium was expected to be present. The aim was to determine the lithium abundance in the different cathode particle types. The same polished blocks used in the AMICS and SEM-EDS analyses were used for the LA-ICP-MS analysis. Six particles were identified and chosen for LA-ICP-MS analyses using spot mode. The associated phases for each of the six particles were provided, whereby the characteristics permitted the precise targeting of the laser to detect lithium and associated metals (Figure 14).

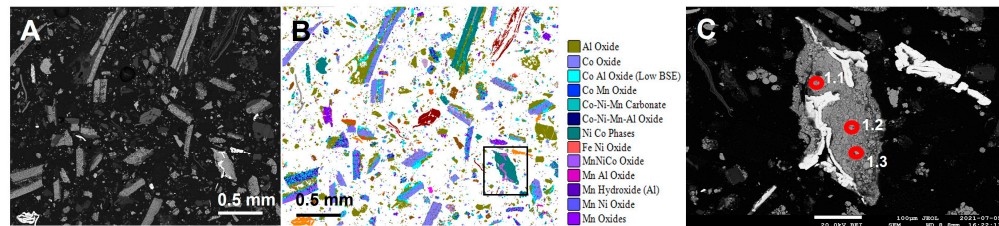

**Figure 14.** Cathode LMO particle, showing (**A**) SEM-BSE and (**B**) automated SEM-EDS images. (**C**) SEM-BSE image; red circles indicate the location of the laser ablation analysis points.

The results of the spot LA-ICP-MS analyses for lithium, cobalt, nickel, copper, zinc, manganese and aluminium are summarised in Table 3. The data for lithium are also summarised in Figure 15. Lithium was detected in all of the particles analysed, although there

was significant variation in the measured lithium abundance. The analyses of Particle 1, assigned to a NiCo phase (AMICS) and interpreted as an NMC cathode particle, had the lowest observed lithium abundance with approximately 1000 ppm. In contrast, Particle 2, assigned to Co Oxide, CoAl Oxide and CoNiMnAl Oxide AMICS compositional groups, and interpreted as an LCO cathode particle, had a lithium abundance ranging between 117,094 and 146,949 ppm (11.7–14.6% lithium). Other analysed particles had lithium abundances ranging between these two. Individual analysed particles typically had internally consistent lithium abundances, which varied significantly between the different particle types.

**Table 3.** LA-ICP-MS analyses (ppm).

| Spot | AMICS Compositional Group | Cathode Type | Li | Co | Ni | Cu | Zn | Mn | Al |
|------|---------------------------|--------------|------|------|------|------|------|------|------|
| 1.1 | NiCo phase | NMC | 1098 | 46,850 | 686,876 | 7204 | 82,375 | 642 | 20,112 |
| 1.2 | NiCo phase | NMC | 1010 | 45,606 | 618,201 | 6529 | 84,578 | 469 | 20,112 |
| 1.3 | NiCo phase | NMC | 1002 | 38,854 | 539,715 | 5760 | 69,695 | 496 | 20,112 |
| 3.2 | NiCo phase | NMC | 197,681 | 154,151 | 175,644 | 188,507 | 3667 | 7753 | 413,995 |
| 4.1 | NiCo Phase, Mn Oxide | NMC | 26,626 | 13,047 | 49,907 | 885 | 28 | 183,738 | 30,167 |
| 4.2 | Mn Oxide | NMC | 25,758 | 14,650 | 59,536 | 2046 | 120 | 244,015 | 30,167 |
| 4.3 | NiCo Phase, Mn Oxide | NMC | 107,953 | 66,389 | 347,350 | 2384 | 175 | 34,716 | 30,167 |
| 4.4 | NiCo Phase, Mn Oxide | NMC | 40,916 | 32,230 | 158,666 | 1410 | 32 | 30,993 | 30,167 |
| 5.1 | Mn Oxide | NMC | 20,745 | 3932 | 9903 | 61 | <1.18 | 218,488 | 3574 |
| 5.2 | Mn Oxide | NMC | 25,455 | 3932 | 15,056 | 33 | 3 | 262,565 | 3974 |
| 5.3 | Mn Oxide | NMC | 72,156 | 3932 | 7795 | 83 | 52 | 660,746 | 6607 |
| 6.1 | NiCo Phase | NMC | 72,996 | 39,596 | 214,262 | 448 | 56 | 752 | 19,053 |
| 6.2 | NiCo Phase | NMC | 62,456 | 31,588 | 183,185 | 449 | 50 | 647 | 19,053 |
| 6.3 | NiCo Phase | NMC | 80,840 | 46,952 | 271,822 | 528 | 53 | 462 | 19,053 |
| 6.4 | NiCo Phase | NMC | 86,717 | 47,026 | 269,028 | 896 | 7 | 607 | 19,053 |
| 2.1 | Co Oxide, CoAl Oxide, CoNiMnAl Oxide | LCO | 137,273 | 202,126 | 189,918 | 11,478 | 40 | 190,843 | 282 |
| 2.2 | Co Oxide, CoAl Oxide, CoNiMnAl Oxide | LCO | 146,949 | 202,126 | 198,121 | 23,029 | 93 | 190,577 | 365 |
| 2.3 | Co Oxide, CoAl Oxide, CoNiMnAl Oxide | LCO | 117,094 | 202,126 | 214,525 | 12,929 | 41 | 187,836 | 2998 |
| 2.4 | Co Oxide, CoAl Oxide, CoNiMnAl Oxide | LCO | 144,534 | 202,126 | 216,398 | 15,113 | 215 | 194,614 | 11,248 |
| 2.5 | Co Oxide, CoAl Oxide, CoNiMnAl Oxide | LCO | 130,095 | 202,126 | 206,958 | 11,103 | 16 | 178,263 | 6750 |
| 2.6 | Co Oxide, CoAl Oxide, CoNiMnAl Oxide | LCO | 136,261 | 202,126 | 206,486 | 25,873 | 157 | 198,542 | 16,837 |
| 2.7 | Co Oxide, CoAl Oxide, CoNiMnAl Oxide | LCO | 132,580 | 202,126 | 191,140 | 4507 | 195 | 204,555 | 840 |

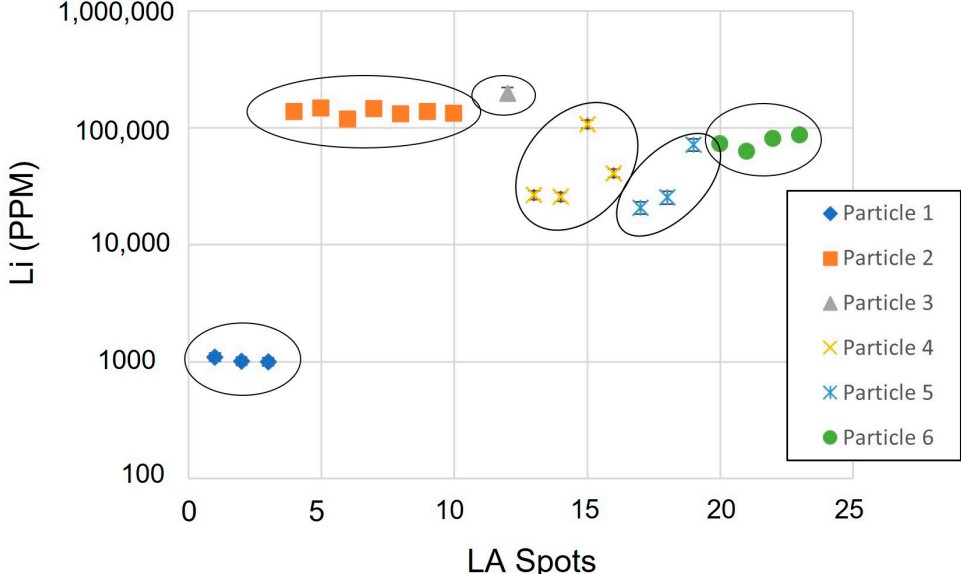

**Figure 15.** Lithium content of each particle measured using LA-ICP-MS.

## 5. Discussion

Some phases identified in the black mass sample could not be attributed back to the original battery components. Further research is therefore recommended to refine the

compositional groups, with the analysis of additional samples. Future analysis should aim to examine unprepared samples and how these change during sample preparation, and in particular the influence of comminution on reducing particle size and liberation.

Importantly, it should be noted that the results of this investigation to determine the phase characteristics of black mass are based on the analysis of a single black mass sample from a European source. It is likely that other samples from different recycling plants (streams) will be variable in composition. Further analysis of a range of different samples is recommended to develop a cost-effective and timely workflow for sample characterisation.

Although the samples analysed using automated SEM-EDS for this project were from a single European source, the initial results show that the cathode materials were compositionally diverse and that the ratio of nickel, manganese and cobalt was especially variable. Qualitative assessment of the EDS spectra indicated that the ratio of these elements was directly linked to the relative peak heights and, therefore, with appropriate reference materials, it should be possible to use spectral matching to further classify the material into specific cathode compositions such as NMC622 ($LiNi_{0.6}Mn_{0.2}Co_{0.2}O_2$), NMC333 ($LiNi_{0.333}Mn_{0.333}Co_{0.333}O_2$), etc. Ultimately, this will allow automated SEM-EDS to characterise the cathode material to a high degree of specificity and provide critical information for the development of effective treatment and recycling workflows.

A scheme to characterise black mass and the developments of standards is important for this emerging and advancing industry, for the following reasons:

- To confirm that a recycled battery material being purchased, sold or traded is actually black mass.
- For the characterisation of the component phases present, prior to sampling and sample preparation.
- To provide quality control measures and to monitor the effectiveness of the preliminary processing stages such as shredding and pyrolysis. Particle and grain size data are important factors in the design and optimisation of grinding circuits.
- To assist in determining the optimal hydrometallurgical and/or pyrometallurgical processing route.
- To detect and quantify payable and penalty phases, which may hinder valuable or precious metal recovery.
- To identify, manage and mitigate hazards and associated levels of risk.
- To assist in determining whether black mass should be regarded as a 'waste' or 'commodity'.

## 6. Conclusions

A programme of innovative and complementary experimental investigations and test work on a single source of black mass from Europe is presented here. Given the different feeds within a recycling plant leading to the formation of the black mass powder for subsequent processing, such powders are likely to be variable in terms of their composition and particle characteristics.

Our results show that automated SEM-EDS can not only characterise and quantify the phases present, but also that the phase chemistry allows the particle types to be assigned back to the primary battery components. Interactive SEM-EDS analysis also demonstrated that there is more detailed higher resolution chemical variability within the individual particle categories.

Micro X-CT imaging helped reveal the 3D form of particles within the analysed sample volume. The 3D particle shape is potentially significant in terms of how the material would behave during subsequent grinding and chemical analysis. Micro-XRF was useful to provide a sample overview and target areas of interest.

Some phases identified are, at present, not attributed back to the original components and further work is needed to refine the compositional groups, with the analysis of additional samples.



Neither automated nor manual interactive SEM-EDS analysis can determine the presence/abundance of lithium. However, the use of LA-ICP-MS was successful in detecting and quantifying the presence of potentially strategically and economically important metals such as lithium.

The combination of analytical methods presented here allowed a considerably better characterisation of the black mass investigated than single methods alone could. From a commercial and operational perspective, the analysis of the black mass powder could be used to: (a) confirm that the material being traded, sold, purchased or processed is actually black mass; (b) characterise the component phases present, prior to processing to (c) assist in determining the optimal processing route; (d) determine the presence of payable phases; (e) identify phases which may hinder metal recovery or may be regarded as penalty components; (f) aid in the development of laboratory assay procedures; and (g) assist in the identification, management and mitigation of hazards and their associated levels of risk.

**Author Contributions:** Conceptualisation and project initiation, L.D.; Methodology, L.D. and D.P.; Sample acquisition and preparation, L.D.; automated mineralogy, D.P. and M.P.; Interactive SEM-EDS analysis, D.P., M.P., S.L. and A.B.; Micro X-CT, Micro XRF and LA-ICP-MS, I.C., J.K., Y.L., X.L., Q.D., E.M.J. and A.B.; Writing—original draft, L.D.; Reviewing and editing, L.D., D.P., M.P., I.C., J.K., S.L., Y.L., X.L., Q.D., E.M.J. and A.B.; Funding acquisition, L.D. All authors have read and agreed to the published version of the manuscript.

**Funding:** This project and investigation into the sampling, sample preparation and phase characterisation of black mass was funded by Alfred H Knight International. X-ray tomography and Micro-XRF were supported by the Academy of Finland via the RAMI infrastructure project (#293109 and #337560, respectively).

**Data Availability Statement:** The data presented in this study are available on request from the corresponding author. The data are not publicly available as AHK Ltd. does not have a web platform to host the data.

**Conflicts of Interest:** LD is an employee of Alfred H. Knight (AHK) Ltd., a company providing inspection services. DP is an employee of the University of South Wales. MRP is an employee at Vidence Inc., a company that provides automated mineral analysis services. I.C., J.K., S.L., Y.L., X.L., Q.D., E.M.J. and A.B. are all employees at the Geological Survey of Finland (GTK), a research organisation governed by the Finnish Ministry of Employment and the Economy, which conducts both non-commercial and commercial activities.

## Appendix A

**Table A1.** Compositional groups and descriptions, automated SEM-EDS analysis.

| Phase | Description |
| --- | --- |
| Carbon | Graphite. |
| Biotite | Biotite and phlogopite mica. Typically occurs in long, expanded flakes. |
| Mg Silicates | Mg silicates such as talc and serpentine. |
| Na Glass | Na-bearing glass compositions. Often occurs as small prills and elongated rod-like particles. May also include other glass compositions (e.g., Ca-bearing glasses, etc.). |
| Other Silicates | Quartz, feldspar and other silicates. May include either naturally occurring mineral matter or anthropogenic material. |
| Calcite and Dolomite | Calcite and dolomite. May include Ca oxides and hydroxides. |
| Apatite | Ca phosphates. |
| Fe Sulphide | Fe sulphides and sulphates. |
| Aluminium | Aluminium metal. |
| Al Carbonate | Carbon with a high Al content. May include aluminium metal finely mixed with carbon. |
| Al Oxide | Aluminium oxides and hydroxides. Typically with a low carbon content. |
| Al K Phosphate | Aluminium oxides and hydroxides containing minor amounts of P and K. Typically has a low carbon content. |

**Table A1.** *Cont.*

| Phase | Description |
|---|---|
| Co Oxide | Granular masses of cobalt oxide. Granules typically less than approximately 15 microns and bound together with an organic binder. Comparatively low BSE indicates that light elements such as Li may also be present. |
| Co Al Oxide (Low BSE) | Aluminium-bearing cobalt oxide. Typically occurs between Co oxide granules, but also occurs as thick, uniform laminae. |
| Co Mn Oxide | Mn-bearing cobalt oxides. Occurs as occasional granules surrounded by an organic binder. |
| Co-Ni-Mn Carbonate | Nickel and cobalt-bearing cobalt oxides and hydroxides. Occurs as granules surrounded by an organic binder. |
| Co-Ni-Mn-Al Oxide_1 | Aluminium-bearing nickel and cobalt-bearing cobalt oxides and hydroxides. Occurs as granules surrounded by an organic binder. |
| Ni Co Phases | Nickel oxides and hydroxides containing small amounts of Co. Often occur as granular masses surrounded by an organic binder. The granules are generally spherical. May also occur as more massive particles. |
| Fe Ni Oxide | Fe oxides and hydroxides containing variable amounts of iron. May include Ni–Fe alloys. |
| MnNiCo Oxide | Similar to Co Ni Mn Oxide but with a slightly lower cobalt content. Occurs as granular masses surrounded by an organic binder. |
| Mn Al Oxide | Manganese-bearing aluminium oxide. High oxygen content but low carbon content. |
| Mn Hydroxide (Al) | Manganese oxide containing minor to trace amounts of Al. |
| Mn Ni Oxide | Nickel-bearing manganese oxides and hydroxides. Often occurs as spherical particles up to ~20 microns in diameter with a radial structure (possibly radiating fibres). |
| Mn Oxides | Mn oxides and hydroxides. Generally occur as aggregates of irregular, angular particles cemented with an organic binder. |
| Fe Al Phosphate | Aluminium-bearing iron phosphate. Occurs as thick, massive laminae up to approximately 100 microns across. Some particles are coated in a layer of aluminium-rich iron phosphate, which has a much lower BSE brightness. |
| Zn Phases | Zinc-bearing phases including zinc oxides, hydroxides and zinc-bearing manganese oxides. |
| Copper | Copper metal. Typically occurs as twisted masses (likely thin, folded sheets). |
| Tin and Cadmium | Cadmium and tin-bearing particles. Most have a high BSE so are likely metal fragments. |
| Ti Phases | Titanium oxides. Fine grained (submicron) granular aggregates are bound by a low BSE (e.g., organic) binder. |
| Fe and Cr Oxides | Iron oxides and hydroxides. Chromium phases are included for brevity. |
| Cl and P | Chlorine and/or phosphorus-bearing materials. Likely organic material or plastics. |
| Other Co Phases | Cobalt-bearing phases not included above. Includes oxides and fluorides. Cobalt typically present only in trace to minor amounts. |
| Other Ni Phases | Nickel phases not included above. Typically includes nickel–REE alloys and oxides. |
| Other Cu Phases | Copper-bearing phases such as copper-aluminium oxides and hydroxides. |
| Unclassified | Particles not reporting to the above categories. May include alloys and hydroxides containing Fe, Co, Ni and Mn in varying amounts. |

## Appendix B

**Table A2.** Modal composition of the samples obtained via automated SEM-EDS analysis.

| Sample Name | AHK/DP/1 | AHK/DP/1 | AHK/DP/1 | AHK/DP/2 | AHK/DP/2 | AHK/DP/2 |
|---|---|---|---|---|---|---|
| **Sample Type** | **Unprepared** | **Unprepared** | **Unprepared** | **Crushed** | **Crushed** | **Crushed** |
| Replicate Number | 1A | 1B | 1A | 2A | 2B | 2B |
| Analysis resolution | 1.59 | 1.59 | 0.83 | 1.59 | 1.59 | 0.83 |
| Carbon (+/−Co Al) | 6.17 | 11.21 | 4.89 | 10.08 | 12.05 | 10.62 |
| Biotite | 1.35 | 1.56 | 0.84 | 1.58 | 1.21 | 1.32 |
| Mg Silicates | 0.03 | 0.09 | 0.03 | 0.17 | 0.06 | 0.06 |
| Na Glass | 0.13 | 0.08 | 0.13 | 0.12 | 0.10 | 0.07 |
| Other Silicates | 0.62 | 0.25 | 0.36 | 0.37 | 0.32 | 0.42 |
| Calcite and Dolomite | 0.04 | 0.07 | 0.05 | 0.08 | 0.11 | 0.03 |
| Apatite | 0.00 | 0.03 | 0.00 | 0.01 | 0.00 | 0.00 |
| Fe Sulphide | 0.02 | 0.00 | 0.02 | 0.00 | 0.00 | 0.00 |

**Table A2.** *Cont.*

| Sample Name | AHK/DP/1 | AHK/DP/1 | AHK/DP/1 | AHK/DP/2 | AHK/DP/2 | AHK/DP/2 |
|---|---|---|---|---|---|---|
| Sample Type | Unprepared | Unprepared | Unprepared | Crushed | Crushed | Crushed |
| Aluminium | 1.07 | 1.00 | 1.08 | 0.69 | 0.70 | 0.84 |
| Al Carbonate | 0.40 | 0.65 | 0.35 | 0.27 | 0.49 | 0.56 |
| Al Oxide | 23.05 | 31.68 | 23.20 | 29.42 | 24.33 | 24.52 |
| Al K Phosphate | 0.94 | 1.36 | 1.05 | 1.43 | 1.15 | 1.11 |
| Co Oxide | 17.08 | 14.46 | 19.97 | 14.33 | 15.49 | 16.45 |
| Co Al Oxide (Low BSE) | 9.52 | 9.42 | 10.12 | 9.78 | 9.84 | 9.83 |
| Co Mn Oxide | 0.05 | 0.06 | 0.04 | 0.07 | 0.06 | 0.05 |
| Co-Ni-Mn Carbonate | 2.01 | 2.00 | 2.09 | 1.94 | 1.60 | 1.78 |
| Co-Ni-Mn-Al Oxide | 2.10 | 2.07 | 1.87 | 2.65 | 2.09 | 2.01 |
| Ni Co Phases | 4.69 | 2.77 | 5.37 | 3.72 | 3.88 | 3.68 |
| Fe Ni Oxide | 0.26 | 0.23 | 0.24 | 0.25 | 0.25 | 0.23 |
| MnNiCo Oxide | 7.10 | 5.16 | 7.74 | 7.27 | 7.21 | 7.94 |
| Mn Al Oxide | 1.32 | 0.94 | 1.20 | 0.98 | 1.61 | 1.59 |
| Mn Hydroxide (Al) | 3.02 | 1.64 | 2.53 | 2.00 | 2.75 | 2.71 |
| Mn Ni Oxide | 0.15 | 0.08 | 0.24 | 0.08 | 0.08 | 0.10 |
| Mn Oxides | 3.85 | 2.13 | 3.40 | 2.95 | 4.06 | 4.11 |
| Fe Al Phosphate | 2.46 | 2.32 | 3.13 | 3.06 | 2.09 | 1.93 |
| Zn Phases | 0.20 | 0.14 | 0.16 | 0.16 | 0.23 | 0.21 |
| Copper | 6.32 | 2.73 | 4.59 | 0.64 | 2.03 | 2.28 |
| Tin and Cadmium | 0.29 | 0.24 | 0.18 | 0.23 | 0.23 | 0.17 |
| Ti Phases | 0.13 | 0.11 | 0.28 | 0.18 | 0.14 | 0.05 |
| Fe and Cr Oxides | 4.50 | 3.70 | 3.91 | 3.68 | 4.11 | 3.86 |
| Cl and P | 0.02 | 0.04 | 0.04 | 0.02 | 0.01 | 0.01 |
| Other Co Phases | 0.01 | 0.01 | 0.01 | 0.00 | 0.02 | 0.05 |
| Other Ni Phases | 0.47 | 0.46 | 0.42 | 0.56 | 0.49 | 0.45 |
| Other Cu Phases | 0.08 | 0.07 | 0.13 | 0.19 | 0.18 | 0.22 |
| Unclassified | 0.56 | 1.22 | 0.38 | 1.03 | 1.04 | 0.77 |
| Total | 100.00 | 100.00 | 100.00 | 100.00 | 100.00 | 100.00 |

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
