# Peer review of "The Recycling of End-of-Life Lithium-Ion Batteries and the Phase Characterisation of Black Mass"

_recycling, doi:10.3390/recycling8040059_

Round 1
Reviewer 1 Report
The reviewed article is self-plagiarism of published work:
The sampling and phase characterisation of black mass, L. Donnelly et al., Proceedings of WCSB10: TOS Forum Issue 11, 397–409 (2022)
Work is, however, mentioned in paper as source [20], which does not justify copying/paraphrasing 2/3 of the volume of the work. Work [20] has been previously reviewed and published, so there is no need to publish the same data multiple times.
In addition, the authors largely use commercially available data analysis system, and evaluation of its work is a large part of the publication. Therefore, it is clearly imitative and does not bring enough new knowledge to the field. Things would have been different if the authors had created new algorithms and data processing methods, but this time nothing of the sort happened.
Author Response
Dear Reviewer 1
Thank you for your review of our paper. Please find attached a response to the points raised.
Best regards
Laurance

Reviewer 2 Report
This work repots the recycling of lithium-ion batteries, focusing on the phase characterisation of black mass. This work is interesting and well designed. Here are some suggestions to improve the manuscript.
In the abstract, the authors have focused on the background and methods of this work. However, the key findings of this work are not well summarized.
Check the language of the whole manuscript, such as line 15: 'battery revolution' and 'green revolution' are likely to ..., line 21: The abundance of light elements such as lithium was..., line 131, etc.
The introduction is too simple and should be strengthened.
The structure of the whole manuscript should be improved. It seems that some sections look like a review article, such as "2. Black Mass", "3. Overview of Analytical Methods", "4. Sampling Standard Operating Procedure (SOP)".
The references are not well distributed throughout the manuscript.
I suggest combining the results and discussion.
The structure of the whole manuscript should be improved.
Author Response
Dear Reviewer
Thank you for your review of our paper. Please find attached a response to the points raised.
Best regards
Laurance

Reviewer 3 Report
Generally, this paper is very interesting. However, some figures are not readable (e.g. 7, 8, 9). It should be improved.
Author Response
Dear Reviewer
Thank you for the review of our paper. Please find attached a response to the points raised.
Best regards
Laurance

Round 2
Reviewer 1 Report
Authors introduced significant corrections to the first version of the work, thanks to which it no longer has the signs of autoplagiarism.
I remain strongly polarized by the first attempt at publication, so I do not make a specific recommendation now. I just want to note that the authors do not indicate their affiliations in the "conflict of interest" section. According to some definition "A conflict of interest can occur when you, or your employer, or sponsor have a financial, commercial, legal, or professional relationship with other organizations, or with the people working with them, that could influence your research." . Representing a commercial company (and not only the scientific sector), which on its website encourages the use of certain analytical techniques, should be indicated in the appropriate section of the work.
Author Response
Responses to referees. We have highlighted the referee's comments and our responses below. We are grateful for the reviewer's ongoing support.
Review 1
Comment 1: Authors introduced significant corrections to the first version of the work, thanks to which it no longer has the signs of autoplagiarism.
Response: We acknowledge the referee's comment.
Comment 2: I remain strongly polarized by the first attempt at publication, so I do not make a specific recommendation now.
Response: We have noted the referee's comment.
Comment 3: I just want to note that the authors do not indicate their affiliations in the "conflict of interest" section. According to some definition "A conflict of interest can occur when you, or your employer, or sponsor have a financial, commercial, legal, or professional relationship with other organizations, or with the people working with them, that could influence your research." Representing a commercial company (and not only the scientific sector), which on its website encourages the use of certain analytical techniques, should be indicated in the appropriate section of the work.
Response: Our interpretation is that the reviewer is concerned that the paper is promoting a methodology, which could financially benefit the organisations of the co-authors, if they are contacted for services to be carried out. We acknowledge and agree with this, and we have therefore added a statement in the manuscript, on page 18 under the sub-heading 'Conflicts of Interest'. The statements is as follows: 'LD is an employee of Alfred H. Knight (AHK) Ltd, a company providing inspection services. DP is an employee of the University of South Wales. MRP is an employee at Vidence Inc, a company provided automated mineral analysis services. IC, JK, SL, YL, XL, QD, EJ and AB are all employees at the Geological Survey of Finland (GTK), a research organization governed by the Finnish Ministry of Employment and the Economy, which conducts both non-commercial and commercial activities.'
Reviewer 2 Report
The revised MS is acceptable.
Minor editing of English language required
Author Response
Responses to referees. We have highlighted the referee's comments and our responses below. We are grateful for the reviewer's ongoing support.
Review 2
Comment: The revised MS is acceptable.
Response: We acknowledge the referee's comment.